# Dissecting Rubella Placental Infection in an In Vitro Trophoblast Model

**DOI:** 10.3390/ijms24097894

**Published:** 2023-04-26

**Authors:** Juliane Schulz, Erik Schilling, Claire Fabian, Ana Claudia Zenclussen, Violeta Stojanovska, Claudia Claus

**Affiliations:** 1Institute of Medical Microbiology and Virology, Medical Faculty, Leipzig University, 04103 Leipzig, Germany; juliane.schulz@bct.uni-halle.de; 2Institute of Biochemistry and Biotechnology, Martin Luther University Halle-Wittenberg, 06120 Halle (Saale), Germany; 3Rheumatology Unit, Department of Internal Medicine III, Medical Faculty, Leipzig University, 04103 Leipzig, Germany; erik.schilling@medizin.uni-leipzig.de; 4Department of Vaccines and Infection Models, Fraunhofer Institute for Cell Therapy and Immunology, 04103 Leipzig, Germany; claire.fabian@izi.fraunhofer.de; 5Medical Department II, University Cancer Center Leipzig (UCCL), University of Leipzig Medical Center, 04103 Leipzig, Germany; 6Department of Environmental Immunology, Helmholtz Centre for Environmental Research, 04318 Leipzig, Germany; ana.zenclussen@ufz.de; 7Perinatal Immunology Research Group, Saxonian Incubator for Clinical Translation, Medical Faculty, Leipzig University, 04103 Leipzig, Germany

**Keywords:** interferon, metabolic activity, extracellular flux analysis, OCR, ECAR, syncytialization

## Abstract

Vertical transmission of rubella virus (RuV) occurs at a high rate during the first trimester of pregnancy. The modes of vertical transmission including the response of trophoblasts to RuV are not well understood. Here, RuV-trophoblast interaction was studied in the BeWo trophoblast cell line. Analysis included early and late time-point kinetics of virus infection rate and the antiviral innate immune response at mRNA and protein level. BeWo characteristics were addressed through metabolic activity by extracellular flux analysis and syncytiotrophoblast formation through incubation with forskolin. We found that RuV infection of BeWo led to profuse type III interferon (IFN) production. Transfecting trophoblast cells with dsRNA analog induced an increase in the production of type I IFN-β and type III IFNs; however, this did not occur in RuV-infected BeWo trophoblasts. IFN-β and to a lesser extent type III IFN-λ1 were inhibitory to RuV. While no significant metabolic alteration was detected, RuV infection reduced the cell number in the monolayer culture in comparison to the mock control and resulted in detached and floating cells. Syncytia formation restricted RuV infection. The use of BeWo as a relevant cell culture model for infection of trophoblasts highlights cytopathogenicity in the absence of a type I IFN response as a pathogenic alteration by RuV.

## 1. Introduction

The placenta forms an effective barrier at the maternal–fetal interface and restricts access of pathogens to the developing fetus through multiple structural, biochemical and innate immunity-based mechanisms [1]. In some cases, maternal infection can be vertically transmitted. Prenatal transmission of bacterial and viral pathogens can occur after infection by members of the TORCH group. This group includes *T*oxoplasmosis, *O*ther (syphilis, varicella-zoster virus, parvovirus B19, Zika virus [ZIKV], human immunodeficiency virus [HIV], malaria), *R*ubella virus (RuV), *C*ytomegalovirus (CMV), and *H*erpes simplex virus [2,3]. A better understanding of placenta morphology, physiology and functionality [4] as well as the mechanisms for vertical transmission of viruses will not only support the identification of therapeutic interventions, but also the evaluation of the potential of emerging pathogens for both placental infections and vertical transmission. The relevance of this is also reflected by the current COVID-19 pandemic, which raised the still prevailing question on the vertical transmissibility of SARS-CoV-2. So far there are only a few reports on SARS-CoV-2 detection in newborns after infection of the mother. This includes a report on an RT-PCR SARS-CoV-2-positive infant born to an asymptomatic SARS-CoV-2-positive woman and one infant born to a SARS-CoV-2-positive mother in a retrospective Polish study on a group of 26 SARS-CoV-2-positive pregnant women who delivered during hospitalization [5,6]. However, positivity could also be due to contact with viral components during labor, e.g., maternal secretion and/or blood might come into contact with the newborn.

Trophoblast cells are central to placental development and function. They are derived from the trophectoderm layer of the blastocyst and can be divided mainly into three populations within the placenta, namely, villous cytotrophoblasts, multinucleated syncytiotrophoblasts and invasive extravillous trophoblasts. Based on their location and their proximity to maternal immune cells, infection can be passed from maternal cells to extravillous trophoblast cells. This would allow for access of pathogens to the villous core and eventually to fetal blood vessels without passage through the restrictive syncytiotrophoblast layer [3,7]. Based on the restriction within the syncytiotrophoblast layer, extravillous trophoblast cells were discussed to be reached by *L. monocytogenes* through cell-to-cell spread from infected maternal phagocytic leukocyte cells [8]. Although *L. monocytogenes* infects extravillous trophoblast cells, the spread in this cell type can be restricted through a reduced rate of listerial vacuolar escape [9]. This aspect emphasizes that there are several layers of restriction of pathogens in the placenta. Another example for the restriction of an infection in the syncytiotrophoblast is CMV, which crosses this layer through antibody-mediated transcytosis of CMV particles that is mediated by Fc receptors expressed on syncytiotrophoblast cells [10].

Viremia during rubella infection in pregnancy can result in placental infection. Early ex vivo studies on tissue samples of 18 second and third trimester placentas derived from pregnancies with maternal rubella infection revealed placentitis in addition to a reduced placental weight, villitis, dysmaturity of villous trees and vascular injury [11]. Additionally, lesions indicative for a cytopathic impact were especially detected in cytotrophoblastic cells [11]. In a further study using one placenta derived after a stillbirth at gestational week 35, RuV antigen was detected in the capillary endothelium and in the basal plate of the placenta [12]. In agreement with this study, Garcia et al. referred to intracytoplasmic inclusion bodies in decidual cells and in cytotrophoblasts [11,12]. Infection of the vascular endothelium was discussed to support viral spread and to result in vascular lesions, while infection of the basal plate leads to impairment of placental functions [12]. Damage of endothelial cells could result in their desquamation to the lumen of blood vessels, which could provide access to the fetal circulation and as such to the fetus [13].

Although RuV can be regarded as an archetypic teratogenic virus for the study of vertical transmission of pathogens, its mode of vertical transmission and the mechanisms of teratogenic alterations during human development are still not completely defined. We hypothesized that trophoblasts as a target cell population for RuV and RuV-induced cytopathogenic alterations can be studied in the BeWo trophoblast cell line. This cell line serves as a model for infection of trophoblasts as well as of syncytialized trophoblasts. With our data, we show that the BeWo cell line supports RuV replication and productive infection. In the presence of type III interferon (IFN) production and the almost total absence of type I IFNs, a substantial cytopathic effect and impairment of the cell monolayer was noted. Finally, we discuss the implication of our data for RuV placental transmission.

## 2. Results

### 2.1. Productive RuV Infection on BeWo after an Initially Low Infection Rate

The reports on the susceptibility of trophoblast cell lines to RuV are divergent. A low susceptibility on first-trimester HTR-8/SVneo and Swan.71 [14] opposes susceptibility analysis of choriocarcinoma cell lines JAR and JEG3 to VSV pseudotyped with RuV structural proteins and a recombinant RuV in a CRISPR/Cas9 genome-wide knockout screening [15,16]. Thus, we aimed at RuV infection mechanisms and antiviral response mechanisms in human choriocarcinoma BeWo cell line. Figure 1A shows a notable infection rate at 72 h post-infection (hpi), which was characterized by a less densely packed cell monolayer as reveled by actin filament counterstaining and by the appearance of several condensed nuclei (Figure 1A). The early and late time-point kinetic analysis revealed an increasing infection rate through virus titer determination (Figure 1B) and viral E1 protein expression through Western blot analysis (Figure 1C,D). In conclusion, BeWo are permissive to RuV and allow a productive infection.

### 2.2. Generation of IFNs during RuV Infection Differs from the Response of BeWo to dsRNA Analog Poly (I:C)

The IFN response is a critical component of the cellular immune response to pathogens and is also active at the placental barrier [17,18]. The transfection of BeWo with the TLR3 agonist and synthetic viral dsRNA analog poly (I:C) induced a sufficient mRNA expression of IFN-β and IFN-λ1 already after 24 h of transfection (Figure 2A). Although at 72 hpi a significant increase in the expression of IFN-β and IFN-λ1 occurred after RuV infection, the fold change especially of IFN-β was reduced in comparison to poly (I:C) transfection (Figure 2B). The secretion of type III IFNs to the supernatant of BeWo at 72 hpi was confirmed by a bead-based multiplex immunoassay (Figure 2C). While IFN-β was detected only for one sample with 197 pg/mL, IFN-λ1 at 1850 ± 1978 pg/mL and IFN-λ2/3 at 8771 ± 10,685 pg/mL were secreted at a significantly increased level compared to the mock control. Next, we addressed IFN secretion at later stages of infection through application of a medium change at 48 hpi as schematically illustrated in Figure 2D. The secretion of type III IFNs between 48 and 72 hpi (Figure 2E) was comparable to the secretion over time of infection without a medium change (Figure 2C). This setting included also one sample of BeWo supernatant with a detectable, but low, amount of IFN-β (Figure 2E). As illustrated in Figure 2F, we transfected RuV-infected BeWo with poly (I:C) as a known trigger for the TLR3-associated innate immune response also active during viral infections. Poly (I:C) was delivered intracellularly through transfection to support its recognition by intracellular dsRNA receptors. The induction of both IFN-β and type III IFN-λ1 and-λ2/3 was verified at protein level in the mock-infected control after transfection of poly (I:C) (Figure 2G). However, in contrast to the transfection of mock-infected BeWo and similar to infection as the sole stimulus (Figure 2C), RuV-infected BeWo trophoblasts were only positive for IFN-λ1 and IFN-λ2/3 after transfection with poly (I:C) (Figure 2H). Mock-infected BeWo generated type I IFN-β at 246 ± 124 pg/mL in response to transfection with poly (I:C), (Figure 2G). The amount of secreted type III IFNs was at 24 h post-transfection with IFN-λ1 at 8182 ± 9213 pg/mL and IFN-λ2/3 at 25,465 ± 8431 pg/mL in mock-infected and poly (I:C)-transfected BeWo (Figure 2G) already higher than in RuV-infected BeWo at 72 hpi (Figure 2C). In conclusion, RuV infection induced especially the production of type III IFNs by BeWo cells, which was not altered by poly (I:C) as a secondary stimulus.

### 2.3. IFNs Interfere with RuV Infection on BeWo

The induction of type I and III IFNs and their antiviral activity occurs not only in a virus-dependent manner, but also in a cell type- and tissue-dependent manner [19]. As a next step we determined the anti-RuV activity of recombinant IFN-β and -λ1. In BeWo cells, IFN-λ1 appears to induce some ISGs at a lower level than IFN-β as representatively shown for interferon-induced transmembrane proteins (IFITM1 and 3) and viperin (Appendix A).

To visualize the effect of the respective IFN treatment, IFN-λ1 was applied at 20 ng/mL as already up to 5 ng/mL were found to be secreted during infection. In relation to IFN-λ1 and based on the higher biological activity of IFN-β, 10 ng/mL were applied either before or after infection or combined. This scheme as illustrated in Figure 3A allowed for assessment of the restriction of RuV from uninfected cells as well as of the impact on the steps of virus replication following virus uptake. Figure 3B shows that only the pre-inf/2 hpi application of IFN-λ1 reduced the release of progeny virus to the medium significantly. However, irrespective of the application time-point, the level of reduction after IFN-β application was higher compared to IFN-λ1. Moreover, after application of IFN-β the number of virus progeny did not or only slightly exceeded the level detected at 6 hpi (Figure 1B), which can be used as a reference for virus particles in the extracellular medium before onset of productive replication. For further analysis of the anti-RuV activity of IFNs, a pre-inf/2 hpi and late-time post-infection (24 hpi) application scheme was followed as representative time points for the highest and lowest impact on virus particle production (Figure 3B). At both application time-points, pre-inf/2 hpi and post (24 hpi)-infection IFN-β significantly reduced intracellular genome copies, while IFN-λ1 had only a minor impact (Figure 3C). This was also reflected in the number of infected cells as semiquantitatively assessed by immunofluorescence analysis (Figure 3D). The number of infected cells after application of IFN-β reduced infection either to a few single cells pre-inf/2 hpi application or to smaller infection centers (24 hpi application), (Figure 3D). Hereafter, semiquantitative immunofluorescence analysis was subjected to counting of C antigen-positive cells. Figure 3E shows that the application of IFN-λ1 either before and after infection or at 24 hpi significantly reduced the number of infected cells at 72 hpi. Either application scheme of IFN-β almost completely abrogated infection (Figure 3E). The antiviral activity of IFN-λ1 was also present at protein level as assessed by Western blot analysis. Here a significant reduction in intracellular viral E1 protein expression was noted after pre-inf/2 hpi-infection application of IFN-λ1 (Figure 3F,G). In conclusion, while exogenous IFN-β had a strong antiviral effect against RuV on BeWo as exemplified for viral genome replication and protein synthesis, viral protein expression was significantly reduced in the presence of IFN-λ1 before and directly after initiation of infection. Both IFN types exerted a stronger antiviral effect following application before and after infection in comparison to the application only after infection.

### 2.4. RuV Infection on BeWo Is Cytopathogenic and Impaired in Syncytialized BeWo

RuV infection of different human and animal epithelial cell lines induced an increase in mitochondrial respiration [20]. Thus, metabolic activity as a relevant reference for cellular functions was determined in BeWo after RuV infection. Figure 4A shows that oxidative consumption rate (OCR) indicative for mitochondrial respiration had a tendency for reduced metabolic activity, but the change in OCR did not reach statistical significance. Extracellular acidification rate (ECAR) indicative for glycolytic activity was almost identical between mock- and RuV-infected cells (Figure 4B). Appendix A shows the measurement file of OCR and ECAR. The change in OCR after injection of FCCP was only modest, but measurable, while a notable increase in ECAR occurred after injection of the metabolic inhibitors. Metabolic potential of mitochondrial respiration was also slightly reduced without statistical significance after RuV infection, while metabolic potential of glycolysis was not affected (Figure 4C). RuV infection of BeWo resulted in cytopathogenic alterations, which were present as detached and floating cells in the supernatant and started already at 48 hpi (Figure 4D). Some recommendations on BeWo subculturing include fresh medium supply as the high glycolytic rate of BeWo likely results in glucose exhaustion (https://www.sigmaaldrich.com/DE/de/product/sigma/cb_86082803, accessed on 14 March 2022). The low OCR/ECAR ratio of mock-infected cells of 1.2 ± 0.3 is in line with this indication of a high glycolytic activity. This ratio is calculated for basal OCR and ECAR and the lower the ratio, the higher is the metabolic reliance of this cell line on glycolysis [21]. In our follow-up analysis of the cytopathogenic impact of RuV infection, we therefore included a medium change at 48 hpi to exclude BeWo impairment based on glucose exhaustion. Figure 4E confirmed the substantial cell loss and impairment of the cell monolayer at 72 hpi. Furthermore, a significant reduction in the number of adherent cells was measured at 72 hpi (Figure 4F). The trypan blue exclusion test revealed a similar rate of less than 10% dead cells for mock- and RuV-infected cells within the adherent monolayer. BeWo cells can be used as a model for fusion of villous trophoblast cells and as such for the syncytiotrophoblast layer [22]. They fuse after the addition of cAMP [23]. The same effect can be replicated by the addition of forskolin to the cell culture medium [24]. We have employed this as shown in Figure 4G to analyze whether RuV can infect syncytialized BeWo. As representatively shown in Figure 4H, viral antigen was not detected in syncytialized BeWo, while surrounding cells were infected at a substantial rate. In conclusion, RuV infection resulted in cytopathogenicity that was characterized by loss of the cell monolayer. Moreover, RuV was excluded from syncytialized BeWo.

## 3. Discussion

We characterized the choriocarcinoma BeWo cell line as an in vitro cell culture model for the RuV infection of placental cytotrophoblast and syncytiotrophoblasts. There are several barriers, including physical and immunological, posed by the placenta to invading pathogens and vertical transmission involves various mechanisms. By using BeWo cells we demonstrate that trophoblasts can be infected and could support persistent infection in the placenta. In contrast to BeWo, HTR8/SVneo cells have a low susceptibility to RuV [14]. Moreover, the use of BeWo as a model for RuV trophoblast infection is beneficial as they are representatives of the villous differentiation pathway of trophoblasts and as such form syncytia after induction of syncytialization with different stimuli [22]. One limitation of this study is that BeWo are an immortalized cell line. Similar to RuV, BeWo were permissive to ZIKV, while HTR8/SVneo were characterized by a low ZIKV infection rate [25]. However, West Nile virus infected both trophoblast cell types at a high rate [25], but is so far not known to be vertically transmitted [26]. Thus, trophoblast infection itself is just one of several factors that contribute to vertical transmission. Nevertheless, trophoblast susceptibility is indicative for placental infection, which is a prerequisite for vertical transmission during pregnancy. The emerging mosquito-borne Rift Valley fever virus (RVFV) causes fatal pregnancy outcomes in livestock animals and miscarriages in humans and infects placental cells of a rat model for congenital infection at a high rate [27]. Moreover, while histopathological placental alterations were noted independently from the teratogenic alterations in the pups, these alterations were more profound and present through multiple placental layers in teratogenic pups [28]. In turn, vertical transmission of SARS-CoV-2 during pregnancy is discussed to be a rare event based on its inefficient replication in the placenta [29].

The syncytiotrophoblast appears to be an effective barrier that limits infections by bacteria, e.g., *L. monocytogenes* [8] and viruses, e.g., CMV [10]. As reviewed by Arora et al., the syncytiotrophoblast layer confers resistance through branched microvilli as parts of the physical restrictions and through a structured and dense cortical actin network [7]. However, this is not a strict exclusion as in mid-gestation human placental tissue, RVFV was also present in the placental syncytial layer [27]. As a follow-up on our observation on the lack of RuV antigen in syncytialized BeWo, future studies should aim at the capacity of RuV-infected trophoblasts to form syncytia and to maintain RuV protein expression after syncytialization. This study just addressed the accessibility of syncytia to RuV through localization of the viral antigen within syncytia. Future studies will address additional markers for syncytia formation including the impact of RuV infection on the production of the human chorion gonadotropin (hCG).

Despite the absence of type I IFN response and significant metabolic alterations, RuV infection of BeWo induced a substantial cytopathic effect and cell loss. This is in line with the literature data on placental alterations after maternal rubella. The ex vivo analysis of placental chorion from maternal rubella cases with therapeutic abortion revealed cellular damage within syncytiotrophoblasts and cytotrophoblasts [13]. Furthermore, in addition to the desquamation of damaged endothelial cells into the fetal circulation, one fatal congenital rubella syndrome case was characterized by damaged cells in the chorion as a source of RuV-carrying diseased cells [13]. This is further substantiated by the detection of necrosis in the decidua after maternal rubella during pregnancy [12] and of a cytopathic effect in cytotrophoblasts [11]. Furthermore, fibrosis was detected in terminal villi [11]. Thus, RuV cytopathogenicity on BeWo appears to be indicative for placental impairment during maternal rubella and differs from other analyzed human and animal cell lines. RuV is characterized by a comparatively slow replication cycle as shown for HUVEC [30] and the highly susceptible African green monkey cell line Vero [31]. RuV usually replicates without the induction of a CPE. However, on susceptible cell lines, such as Hs888Lu as a human adult lung fibroblast cell line in addition to animal Vero and RK13 (a rabbit kidney cell line), a CPE composed of cell rounding and detachment can be noted [32,33]. However, even on susceptible cell lines, some clinical isolates lack CPE induction [34].

In contrast to BeWo, endothelial HUVEC, lung carcinoma A549 and PBMC-derived macrophages respond to rubella infection with the generation of type I IFN-β [35,36]. For most of the samples analyzed, no IFN-β was detected at protein level. Protein degradation could happen over time of incubation, especially in the case of a low-level production. The pattern recognition receptor (PRR) for RuV is still ill defined. A recent publication revealed that the melanoma differentiation-associated gene 5 (MDA5) as a member of the retinoic acid-inducible gene I (RIG-I)-like receptors (RLRs) appears to be involved in the recognition of RuV in human neuronal cells [37]. MDA5 also recognizes poly (I:C) as a long dsRNA structure [38]. Toll-like receptor 3 (TLR3) as another PRR for poly (I:C) is expressed in BeWo [39]. However, in the presence of RuV the induction of IFN-β at mRNA level was also comparatively low in comparison to the transfection of poly (I:C). The RuV infection rate was lower than 40%, which might also result in an inconsistent IFN response and a differential response compared to poly (I:C). The presence of PRRs as well as the levels of IFN expression during placental maturation are under discussion; IFN expression levels were higher in mid-gestational than in early-gestational chorionic villus tissue [40]. Thus, the antiviral response to RuV might be different during placental development, which could be addressed through primary trophoblast cells. As shown here, BeWo respond to poly (I:C) transfection with the production of type I IFNs, but not as a secondary challenge to a prior RuV infection. This could indicate some interference by RuV with type I IFN synthesis. Accordingly, the level of IFN expression after infection of human villous trophoblasts with RVFV was influenced by the presence of the viral nonstructural protein from the S gene (NSs protein) as a well-known IFN antagonist [41]. The fold increase in especially type I, but also type III, IFN expression was significantly higher after infection with a recombinant NSs-deleted strain compared to the wild-type strain [41]. Although IFN production after a medium change of RuV-infected human PBMC-derived macrophages was diminished, it was restored after a secondary challenge with LPS [36]. In contrast to this, a challenge of RuV-infected BeWo with poly (I:C) induced IFNs comparable to the transfection control and to a medium change in the absence of transfection. Future research needs to evaluate how IFNs exert antiviral activity against RuV infection and how RuV could potentially interfere with IFN production as indicated by both the analysis of infected macrophages and trophoblasts. Albeit at a lower level compared to IFN-β, IFN-λ1 was active against RuV. The reduced antiviral activity of IFN-λ1 against RuV on BeWo might be due to a differential expression of ISGs or their different kinetics. Follow-up studies are required to address how the differential expression of ISGs relates to the antiviral activity of IFNs against RuV. As shown for human mid-gestation chorionic villous explants, recombinant IFN-β, but not IFN-λ3, induced morphological alterations indicative for damage to placental villi and subsequently for placental dysfunction [42]. Thus, the type of IFN produced during infection has implications for viral pathology in the placenta.

RuV infection of BeWo cells was accompanied by a trend towards a reduced mitochondrial respiration. Although not statistically significant, this observation is noteworthy as in epithelial lung carcinoma A549 cells an increase in stressed OCR after injection of respiratory inhibitors was detected [43]. Similar to RuV, ZIKV infection of primary placental cells reduced OCR as indicative for mitochondrial respiration, while glycolysis was not altered [44]. As a mechanism discussed, the reduction in mitochondrial respiration during ZIKV infection of placental cells could favor lipid accumulation in placental tissues through reduced β-oxidation and as such lipid metabolism [44]. The tendency of a reduced mitochondrial respiration could be further addressed in future studies that employ an uncoupler different from FCCP. Easton et al. reported that progesterone released by BeWo could recouple mitochondria that were uncoupled by FCCP, which was not observed with the mitochondrial uncoupler dinitrophenol (DNP) [45]. Thus, future extracellular measurements of the metabolic activity in RuV-infected BeWo should be focused on mitochondrial respiration and employ different uncouplers aside from FCCP.

Vertical transmission of pathogens comprises both replication in placental cell populations such as trophoblasts and the ability to cross the placenta and to reach the fetal system. Here, we studied RuV infection in BeWo as a representative cell type for the cytotrophoblast differentiation pathway. With our data we provide important aspects for the future evaluation of RuV placental infection, impact of syncytialization, and differential modulation of viral transfer across the placenta.

## 4. Materials and Methods

### 4.1. Cell Culture and Virus Infection

BeWo cells (ATCC, CCL-98) were maintained in Dulbecco’s Modified Eagle Medium F-12 Nutrient Mixture (DMEM, F-12) with GlutaMAX (#31331-028, Thermo Fisher Scientific, Gibco Life Technologies, Waltham, MA, USA), 10% fetal bovine serum (FBS, #10437-028, Thermo Fisher Scientific, Gibco Life Technologies) and 10,000 U/mL penicillin and 10,000 µg/mL streptomycin. Based on their high glycolytic rate, a medium change or passaging through trypsinization was performed every 48 h. BeWo cells were plated at 8 × 10^4^ cells per well of a 24-well plate 24 h prior to infection. Virus infections were carried out at a multiplicity of infection (MOI) of 5 in a low volume of DMEM-F12 with 1% FBS. The RuV clinical isolate RVi/Wuerzburg.DEU/47.11 (Wb12, genotype 2B) was used. Virus stock preparation and titer determination by standard plaque assay were carried out on Vero cells (ATCC, CCL-81). The Vero cell line was cultivated in DMEM with high glucose and GlutaMAX (#61965-026, Thermo Fisher Scientific, Gibco Life Technologies), 10% FBS and 10,000 U/mL penicillin and 10,000 µg/mL streptomycin. A humidified incubator with 5% CO_2_ atmosphere was used for cell cultivation at 37 °C.

### 4.2. Application of Interferons

In the exogenous application of human IFNs, IFN-β (10 ng/mL; 300-02BC, Peprotech, Neuilly-sur-Seine, France) and IFN-λ1 (20 ng/mL; #300-02L, Peprotech) were used. IFNs were added either 3 h before infection (pre-infection [pre-inf]), 2 h (2 hpi) or 24 h (24 hpi) after infection. In the pre-inf approach, IFNs remained on the cells until the medium was changed in the course of the infection. In the continuous IFN treatment approach, pre-inf and 2 hpi additions were combined.

### 4.3. Extracellular Flux Analysis

Metabolic analysis was performed as a real-time measurement through extracellular flux analysis with XFp extracellular flux analyzer (Agilent Seahorse Technologies, Santa Clara, CA, USA). The oxygen consumption rate (OCR) and extracellular acidification rate (ECAR) were measured with XFp flux analyzer through application of the cell energy phenotype test kit (Agilent Seahorse Technologies) according to the manufacturer’s instructions. As metabolic inhibitors, the ATP synthase inhibitor oligomycin (1 µM) was used as a co-injection with trifluoromethoxy carbonylcyanide phenylhydrazone (FCCP), (1 µM) after 3 basal measurement points. BeWo were plated in XFp miniplates at a density of 0.5 to 1 × 10^4^ cells per well. Measurements were performed in biological and experimental triplicates. A medium change prior to measurement was conducted with XF DMEM (#103575-100, Agilent Seahorse Technologies) under the addition of 10 mM glucose, 1 mM sodium pyruvate and 2 mM L-glutamine. Normalization of OCR and ECAR values was performed through total protein content determined as optical density (OD) by Bradford assay.

### 4.4. Induction of Syncytialization

For induction of syncytialization, 5000 cells were plated per well of a 24-well plate. After an incubation period of 24 h, syncytialization was induced through the addition of 50 µM forskolin (#SC-3562 Santa Cruz Biotechnology, Dallas, TX, USA) and a total incubation time of 72 h. Forskolin was dissolved as a 10 mM stock solution in 50% (*v/v*) DMSO in PBS, stored at 4 °C and added daily with fresh maintenance medium.

### 4.5. RNA Extraction

Total cellular RNA was extracted at indicated time-points by the ReliaPrepTM RNA cell miniprep system (#Z6011, Promega, Madison, WI, USA) based on lysis with 1-thioglycerol as instructed by the manufacturer.

### 4.6. Cellular mRNA Expression

Reverse transcription was performed with 600 ng of total RNA and oligo (dT_18_) and AMV reverse transcriptase (Promega). For qRT-PCR a 1:5 dilution of cDNA samples was used with the GoTaq^®^ qPCR master mix (#A6001, Promega). For normalization of the relative expression of indicated genes, the 2^−ΔΔCt^ method with the β-actin gene was employed. Sequences of oligonucleotides for indicated genes were as follows with respective annealing temperature in brackets: β-actin forward CCTTCCTGGGCATGGAGTCCTG and reverse GGAGCAATGATCTTGATCTTC (57 °C); IFN-β forward GCCGCATTGACCATCTAT and IFN-β reverse GTCTCATTCCAGCCAGTG (60 °C) [46]; IFN-λ1 forward GCAGGTTCAAATCTCTGTCACC and reverse AAGACAGGAGAGCTGCAACTC (60 °C) [47]; IFITM1 forward CCAAGGTCCACCGTGATTAAC and reverse ACCAGTTCAAGAAGAGGGTGTT (57 °C) [48]; IFITM3 forward GATGTGGATCACGGTGGAC and reverse AGATGCTCAAGGAGGAGCAC (57 °C) [49]; viperin GAGAGCCATTTCTTCAAGACC and CTATAATCCCTACACCACCTCC (60 °C).

### 4.7. Viral Genome Quantification

One-step TaqMan reverse transcription quantitative PCR was performed as described [50]. For amplification of a p90 gene region sense primer (RV_235.s, 5′-CTGCACGAGATYCAGGCCAAACT-3′) and antisense primer (RV_419.as, 5′-ACGCAGATCACCTCCGCGGT-3′) were employed together with a TaqMan fluorogenic probe (RV_291TaqFAM, 6FAM-TCAAGAACGCCGCCACCTACGAGC-BBQ). For reverse transcription, the QuantiTect Probe Reverse Transcription kit (#204443, QIAGEN, Hilden, Germany) was used.

### 4.8. Transfection of dsRNA Analog Poly (I:C)

For transfection with polyinosinic-polycytidylic acid [poly (I:C)], (#sc-202767, Santa Cruz Biotechnology), 7 × 10^4^ BeWo were seeded per well of a 24-well plate and were either transfected 24 h after seeding or at 48 hpi. Before transfection a medium change to 500 µL of maintenance medium (DMEM-F12 with GlutaMAX and 10% FBS) was performed. For transfection 1.5 µL of Lipofectamine 3000 (Thermo Fisher Scientific, Invitrogen, Waltham, MA, USA) were added to 25 µL of serum-free medium. In a separate reaction tube, 500 ng of poly (I:C) were added to 25 µL of serum-free medium. The poly (I:C) mixture was added to the Lipofectamine 3000 mixture. After an incubation for 5 min at room temperature, the Lipofectamine 3000-poly (I:C) mixture was added to the respective wells and incubated for 24 h. As solvent control (CTRL), a Lipofectamine-mixture was employed.

### 4.9. Assessment of Adherent Cell Monolayer

A 0.5% crystal violet stain with a 4% formaldehyde solution in PBS was used for staining of adherent cells.

### 4.10. Immunofluorescence Analysis

BeWo seeded on glass coverslips were fixed at 72 hpi with 4% formaldehyde (Roti^®^-Histofix, #P087, Roth, Karlsruhe, Germany) for 1 h at room temperature. After two PBS washing steps, cells were permeabilized and blocked with 0.3% Triton X-100 in PBS (*v/v*) and 5% normal donkey and goat serum (Dianova, Geneva, Switzerland) (*v/v*) in PBS for 30 min at 37 °C. Thereafter, cells were incubated with a 1:200 dilution of anti-capsid protein (C) antibody (clone 2-36, Meridian Life Science, Cincinnati, OH, USA) as primary antibody for 60 min at 37 °C and polyclonal anti-E-cadherin rabbit antibody (#20874-1-AP, Proteintech, Rosemont, IL, USA) at a 1:200 dilution. This was followed by a 1:200 dilution of Cy3-conjugated donkey anti-mouse IgG as secondary and Alexa488 goat anti-rabbit antibody (Dianova, Hamburg, Germany) for 45 min. Three washing steps with PBS were performed after incubation with primary and secondary antibodies. Nuclei were counterstained during mounting with fluoromount-G™ (#00-4959-52, Thermo Fisher Scientific, Invitrogen). Labelling of actin filaments was performed with Alexa488 phalloidin (Thermo Fisher Scientific, Invitrogen) at a 1:40 dilution for 30 min at room temperature after the blocking and permeabilization step. Primary anti-C antibody was added after two PBS washing steps. Microscopic fluorescence images were obtained with the Olympus IX73 (Evident, Olympus, Tokyo, Japan). ImageJ 1.53k was used for quantification of immunofluorescence images. Two to three images of random microscopic fields were used per sample for counting of RuV C antigen-positive cells as % of total cells.

### 4.11. Bead-Based Multiplex Assay for IFN Quantification in Cell Culture Supernatants

The LEGENDplex human type 1/2/3 interferon panel (#740396, BioLegend, San Diego, CA, USA) was used for quantification of undiluted cell culture supernatants as indicated by the manufacturer. The cytometer MACSQuant X (Miltenyi Biotec, Bergisch Gladbach, Germany) was used for sample assessment.

### 4.12. Western Blot Analysis

Total BeWo cell extracts were prepared by RIPA buffer cell lysis and 30 µg of proteins were separated by SDS-Page and transferred to nitrocellulose membrane for immunoblot analysis with anti-GAPDH antibodies (#sc32233, Santa Cruz Biotechnology, 1:1000 dilution) and anti-rubella E1 antibody (clone EI-2) antibodies (1:500, # MAB925, Merck Chemicals). Chemiluminescence imaging with ECL Prime Western Blotting Reagents and ECL start Western Blotting detection reagents (Amersham) was recorded on Imaging System Azure 300. Relative protein expression was calculated with ImageJ 1.53k.

### 4.13. Statistics

Data in the diagrams are shown as mean + standard deviation (SD) for n = 3 to 5 samples. Three experimental replicates were employed in extracellular flux analysis. Statistical significance was calculated using GraphPad Prism software 9 (San Diego, CA, USA). Unpaired *t*-tests as well as one-way ANOVA tests with Tukey’s or Dunnett’s multiple comparison were used as indicated. Significance in comparison to the mock or solvent control was given with a *p*-value < 0.05 and is indicated as *.

## Figures and Tables

**Figure 1 ijms-24-07894-f001:**
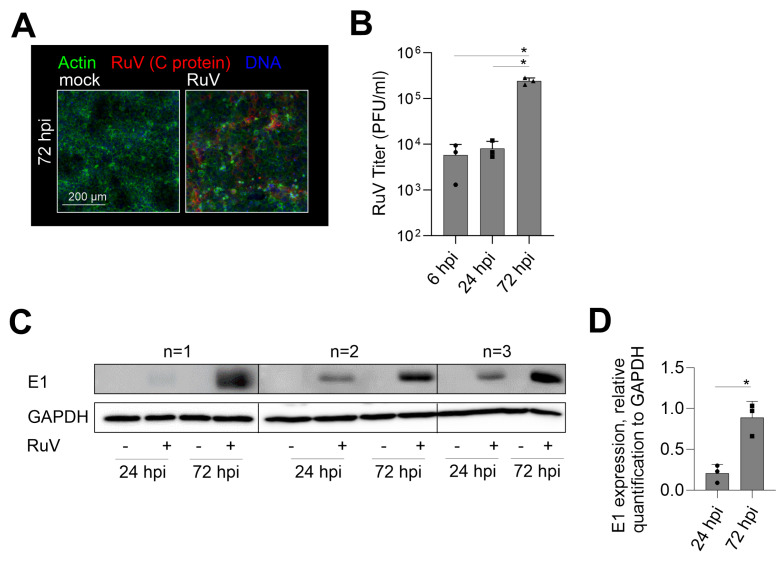
RuV productively infects BeWo cells at an increasing rate over time of incubation. (**A**) Representative images of the immunofluorescence analysis of BeWo at 72 hpi with anti-C protein antibodies (viral antigen) together with counterstaining of actin filaments with Alexa Fluor 488 phalloidin and of DNA with DAPI. Scale bar, 200 µm. (**B**) Virus titer as indicative for virus particle production over time of infection was determined by plaque assay. (**C**) Western blot analysis of RIPA cell lysates of mock- and RuV-infected BeWo at 72 hpi was carried out with antibodies directed against viral E1 protein and GAPDH. (**D**) Densitometric analysis of E1 expression relative to the expression of GAPDH. (**B**,**D**) Data are shown as mean + SD; n = 3; * *p* < 0.05 as determined by (**B**) one-way ANOVA with Tukey’s correction and (**D**) unpaired *t*-test.

**Figure 2 ijms-24-07894-f002:**
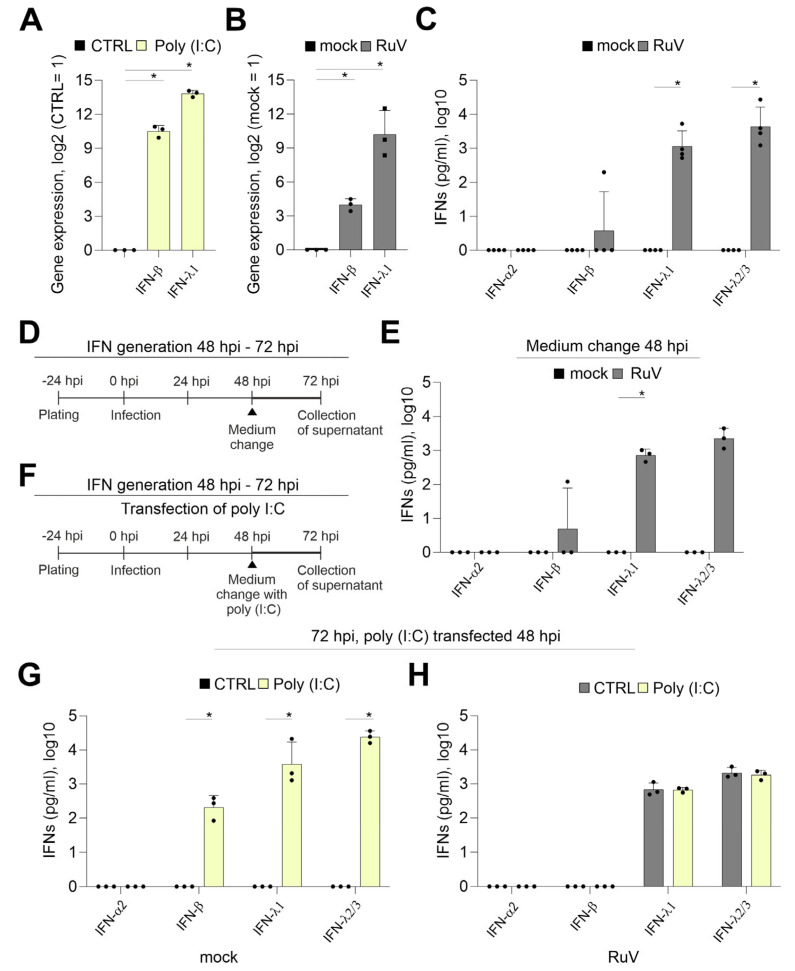
In contrast to transfection with the viral dsRNA analog poly (I:C), the response of BeWo to infection with RuV is mainly based on type III IFNs. Quantification of mRNA expression of indicated IFNs (**A**) at 24 h after transfection of BeWo with 500 ng poly (I:C) per well and (**B**) at 72 hpi after infection with RuV by qPCR, normalized to β-actin mRNA in the comparative cycle threshold method (ΔΔCt). (**C**,**E**,**G**,**H**) Secretion of IFNs into the supernatant of treated BeWo was analyzed by bead-based multiplex LEGENDplex human IFN panel. (**D**) and (**F**) Schematical illustration of the experimental conditions for (**E**) and for (**G**,**H**), respectively. LEGENDplex-based measurement of (**C**) mock- and RuV-infected BeWo at 72 hpi, (**E**) mock- and RuV-infected BeWo at 72 hpi after medium change performed at 48 hpi and (**G**) mock-infected BeWo at 72 hpi after transfection at 48 hpi with 500 ng poly (I:C) per well and (**H**) RuV-infected BeWo at 72 hpi after transfection at 48 hpi with 500 ng poly (I:C) per well. All data (n = 3 to 4) are shown as mean + SD, * *p* < 0.05 as determined by unpaired Student’s *t*-test.

**Figure 3 ijms-24-07894-f003:**
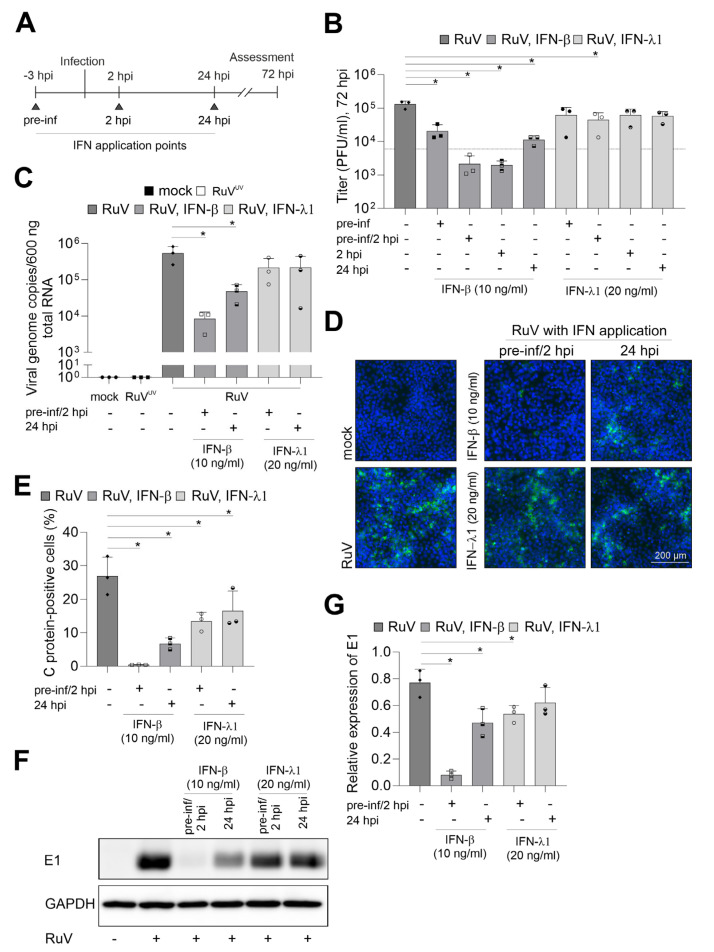
IFN-β efficiently blocks RuV infection in BeWo. (**A**) Graphical illustration of the application of IFNs in reference to the time-point of infection. Triangles indicate IFN application time points. (**B**) Virus titer in the presence and absence of IFNs was determined by plaque assay. As a reference for progeny virus before productive infection, a dashed line corresponding to the particle titer at 6 hpi was added. (**C**) RuV genome copies in mock- and RuV-infected samples were determined at 72 hpi by one-step TaqMan RT-PCR. UV-inactivated RuV was used as a control. Type I and III IFNs were added before and after infection as indicated. (**D**) Immunofluorescence with anti-C protein antibodies was carried out at 72 hpi for indicated samples. DNA was counterstained with DAPI. (**E**) Fluorescence microscopy images were used for quantification of the number of C protein-positive cells as percentage of total cells. (**F**) Western blot analysis of RIPA cell lysates of mock- and RuV-infected BeWo at 72 hpi with antibodies directed against viral E1 protein and GAPDH. (**G**) Densitometric analysis of viral E1 expression relative to the expression of GAPDH. (**B**,**C**,**E**,**G**) data (n = 3) are shown as mean + SD; n = 3; * *p* < 0.05 from one-way ANOVA with Dunnett’s correction.

**Figure 4 ijms-24-07894-f004:**
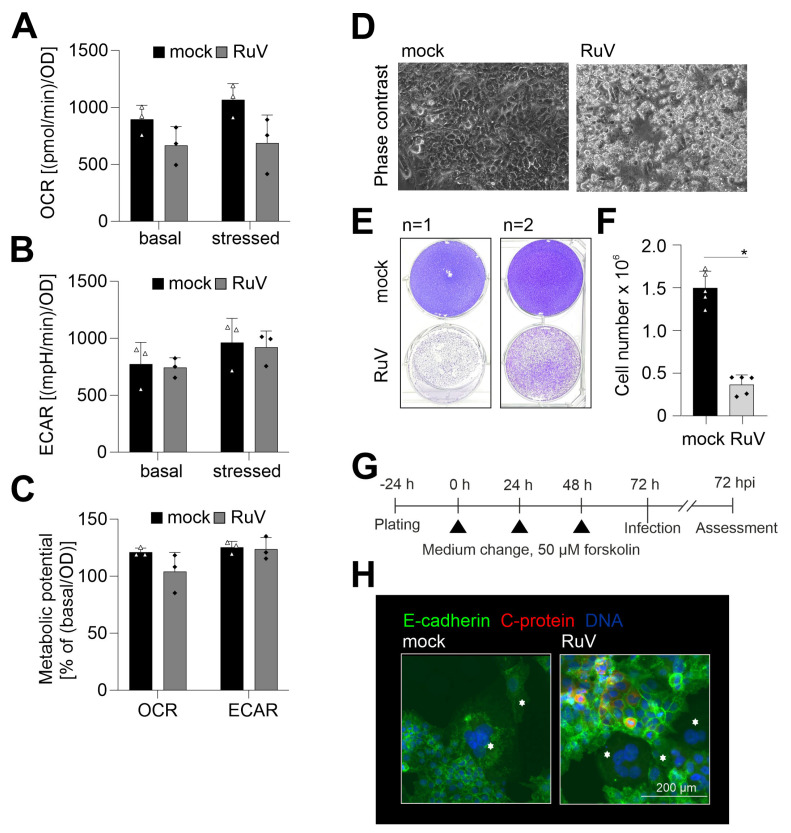
Along with a slight reduction in metabolic activity, RuV infection induced a cytopathic effect with cell loss and did not infect syncytialized cells. (**A**–**C**) Metabolic activity in mock- and RuV-infected BeWo was determined at 72 hpi by extracellular flux measurement with the cell energy phenotype kit. The energy phenotype test report generator software was used for quantification of (**A**) OCR as indicative for mitochondrial respiration and (**B**) ECAR as indicative for glycolysis. Values are shown as basal metabolic activity and stressed metabolic activity (after co-injection of inhibitors of mitochondrial respiration). (**C**) Basal and stressed OCR and ECAR values were used for calculation of the metabolic potential. (The metabolic potential is based on the percent increase in stressed OCR and ECAR in comparison to basal OCR and ECAR, respectively). (**D**) Representative phase contrast microscopic images of mock- and RuV-infected BeWo at 48 hpi. Scale bar, 200 μm. (**E**) Staining of the adherent cell layer of mock- and RuV-infected BeWo at 72 hpi in a 6-well plate with crystal violet dye. Shown are two representative images out of four independent experiments. (**F**) Cell layer of mock- and RuV-infected BeWo was trypsinized at 72 hpi and counted with a hemocytometer. (**G**) Graphical scheme for the induction of syncytialization through medium change with the application of 50 µM forskolin followed by infection and further incubation for 72 h. (**H**) BeWo were incubated with 50 µM forskolin for 72 h followed by mock- and RuV-infection and fixation at 72 hpi. Representative image of the immunofluorescence analysis of mock and virus infection in syncytialized BeWo with anti-E-cadherin as a reference for surface staining and anti-C protein antibodies for visualization of viral antigen. Syncytia are marked with asterisks. (**A**–**C**,**F**) data (n = 3 to 5) are shown as mean + SD, * *p* < 0.05 as determined by unpaired Student’s *t*-test.

## Data Availability

Not applicable.

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
