# Peer review of "Dissecting Rubella Placental Infection in an In Vitro Trophoblast Model"

_ijms, 2023, doi:10.3390/ijms24097894_

Round 1

Reviewer 1 Report

The manuscript by Schulz et al. focuses on investigating the effect of rubella virus (RuV) infection in the BeWo trophoblast cell line. The authors show that BeWo cells are susceptible to RuV infection and that these cells respond to the viral infection by mainly producing IFN-III. Ultimately, this infection is cytopathic; however, RuV does not seem to infect syncytialized cells. The manuscript is well written, and the results provide exciting proof of concept regarding RuV infection of BeWo trophoblasts.

Major comments: 

  1. Is it known which PRRs recognize RuV? Do BeWo cells express TLR3? Because the transfection was delivered intracellularly and TLR3 is usually expressed in endosomes, what was the rationale for using the TLR3 agonist dsRNA analog poly (I:C) as a positive control for IFN-I production? Since RuV is an ssRNA virus, why did the authors use poly (I:C) and not TLR7/8 or RIG-I/MDA5 agonists instead or additionally, which should mimic the ssRNA viral structure more similarly than poly (I:C).
  2. The authors report the number of infected cells using immunofluorescence in Figure 3D, and although visually, the images look different, no quantification is shown for this experiment. Can the authors quantify the number of infected cells in each condition? 

Author Response

The authors want to thank the reviewer for the time spent on our manuscript and the helpful comments and suggestions. The comments were addressed as follows:

(1) Is it known which PRRs recognize RuV? Do BeWo cells express TLR3? Because the transfection was delivered intracellularly and TLR3 is usually expressed in endosomes, what was the rationale for using the TLR3 agonist dsRNA analog poly (I:C) as a positive control for IFN-I production? Since RuV is an ssRNA virus, why did the authors use poly (I:C) and not TLR7/8 or RIG-I/MDA5 agonists instead or additionally, which should mimic the ssRNA viral structure more similarly than poly (I:C).

# We have added literature data on the involvement of MDA5 in the recognition of RuV (Sakuragi et al., 2022). Poly (I:C) as utilized in our study is recognized by both, MDA5 and TLR3. We have also added literature data on the expression of TLR3 in BeWo (Gierman et al., 2015).

Accordingly, line 460 to 465, page 14 states: “The pattern recognition receptor (PRR) for RuV is still ill defined. A recent publication revealed that melanoma differentiation-associated gene 5 (MDA5) as a member of the retinoic acid-inducible gene I (RIG-I)-like receptors (RLRs) appears to be involved in the recognition of RuV in human neuronal cells [3]. MDA5 also recognizes poly (I:C) as a long dsRNA structure [4]. Toll-like receptor 3 (TLR3) as another PRR for poly (I:C) is expressed in BeWo [5].”

(2) The authors report the number of infected cells using immunofluorescence in Figure 3D, and although visually, the images look different, no quantification is shown for this experiment. Can the authors quantify the number of infected cells in each condition? 

#Thank you for this suggestion, which added valuable information to our manuscript.

The data set as suggested was derived from the quantification of fluorescence microscopy images as described on page 5, line 199 to 201 and was added to Figure 3 as (E). In the results section we added: line 319 to 323, page 8: “Hereafter semiquantitative immunofluorescence analysis was subjected to counting of C antigen-positive cells. Figure 3E shows that the application of IFN-λ1 either before and after infection or at 24 hpi significantly reduced the number of infected cells at 72 hpi. Either application scheme of IFN-β almost completely abrogated infection (Figure 3E).” and line 329 to 330, page 9: “Both IFN types exerted a stronger antiviral effect following application before infection”. Additionally, we also included the potential association of infection rate with IFN response: line 467 to 468, page 14: “RuV infection rate was lower than 40%, which might also result in an inconsistent IFN response and a differential response compared to poly (I:C).”

Reviewer 2 Report

This is an interesting work dissecting rubella placental infection in an in vitro model using BeWo cells. Forskolin was used to create syncytiotrophoblast formation. The authors found that rubella infection of BeWo cells led to profuse type III interferon production. In addition, the author reported that syncytia formation restricted rubella infection. Consequently, this work’s findings led to a suggestion of using BeWo as a relevant cell culture model for the infection of trophoblasts.

 The manuscript is quite well written.

Author Response

This is an interesting work dissecting rubella placental infection in an in vitro model using BeWo cells. Forskolin was used to create syncytiotrophoblast formation. The authors found that rubella infection of BeWo cells led to profuse type III interferon production. In addition, the author reported that syncytia formation restricted rubella infection. Consequently, this work’s findings led to a suggestion of using BeWo as a relevant cell culture model for the infection of trophoblasts.

 The manuscript is quite well written

#The authors are very grateful for this positive comment and for the time spent on the review of our manuscript.

Reviewer 3 Report

To test the usefulness of BeWo cells, a placental trophoblast cell line, as a model for placental infection with rubella virus, Schulz et. al. analyzed the growth of rubella virus in these cells and the response of the cells to the infection. They have shown that rubella virus productively infects BeWo cells and induces cell death and Type I and III interferon responses. They also have shown that secondary interferon stimulation by polyI:C does not occur in rubella virus-infected BeWo cells. Furthermore, they have shown that rubella virus does not infect syncytioal BeWo cells induced with forskolin.

Because no reliable animal model of transplacental transmission of rubella virus has been established, analysis of this virus infection in placental trophoblast cells will provide important information for understanding the mechanism of transplacental transmission of this virus. Especially, the kinetics of interferon production and cellular metabolism of BeWo cells during rubella virus infection seems to be an important insight into the pathogenicity. In addition, the fact that induced syncytiotrophoblasts are resistant to rubella virus infection is a very interesting finding for elucidating the route of transmission to the fetus.

Because several points are unclear, the following comments should be considered and the manuscript should be revised.

Comments

1)                  Fig.4H: The authors state that "viral antigen was not detected in syncytialized BeWo," but the results of Fig. 4H alone are not clear. Quantitative data is required. And, it would be informative to detect antigens specific to syncytiotrophoblasts, such as hCG, in order to determine whether these cells exhibit a feature of syncytiotrophoblasts.

2)                  An interesting point in this manuscript is that RuV induces Type 3 IFNs in BeWo cells, but Type 3 IFNs have weak RuV-suppressive capacity. Conversely, IFN-β is not strongly induced by RuV infection, but has high anti-RuV activity. Unfortunately, there is no discussion in the manuscript about the difference in rubella virus-suppressive activity between Type 1 IFNs and Type 3 IFNs. Please discuss this point. An examination of whether there is a difference in the induction of ISG when each IFN is added to BeWo cells might provide a very useful for understanding it.

Author Response

The authors want to thank the reviewer for the time spent on our manuscript and the helpful comments and suggestions. The comments were addressed as follows.

1)Fig.4H: The authors state that "viral antigen was not detected in syncytialized BeWo," but the results of Fig. 4H alone are not clear. Quantitative data is required. And, it would be informative to detect antigens specific to syncytiotrophoblasts, such as hCG, in order to determine whether these cells exhibit a feature of syncytiotrophoblasts.

To clarify this raised concern, we added the limitation of the current study to the discussion section: line 434 to 437, page 13: “This study just addressed the accessibility of syncytia to RuV through localization of viral antigen within syncytia. Future study will address additional markers for syncytia formation including the impact of RuV infection on the production of the human chorion gonadotropin (hCG).” Additionally, we need to evaluate whether infection itself has an impact on hCG production after syncytialization.

2) An interesting point in this manuscript is that RuV induces Type 3 IFNs in BeWo cells, but Type 3 IFNs have weak RuV-suppressive capacity. Conversely, IFN-β is not strongly induced by RuV infection, but has high anti-RuV activity. Unfortunately, there is no discussion in the manuscript about the difference in rubella virus-suppressive activity between Type 1 IFNs and Type 3 IFNs. Please discuss this point. An examination of whether there is a difference in the induction of ISG when each IFN is added to BeWo cells might provide a very useful for understanding it.

#Thank you for this suggestion. We have addressed mRNA expression level of three representative ISGs (IFITM1, IFITM3, viperin) by quantitative PCR and added the obtained data set as Supplement Figure 1. In the results section we included the following statement: line 293 to 296, page 8: “In Bewo cells, IFN-λ1 appears to induce some ISGs at a lower level than IFN-β as representatively shown for interferon-induced transmembrane proteins (IFITM1 and 3) and viperin (Supplement Figure 1).” Additionally, we also included this aspect in the discussion section: line 486 to 490, page 14: “Albeit at a lower level compared to IFN-β, IFN-λ1 was active against RuV. The reduced antiviral activity of IFN-λ1 against RuV on BeWo might be due to a differential expression of ISGs or their different kinetics. Follow-up studies are required to address how differential expression of ISGs relates to the antiviral activity of IFNs against RuV.“ Primer sets used were added to the methods section, page 4, line 158 to 161.